# Turbo Connection: Reasoning as Information Flow from Higher to Lower Layers

**Mohan Tang** [1]   **Sidi Lu** [1]

## Abstract

Complex problems, whether in math, logic, or planning, are solved by humans through a sequence of steps where the result of one step informs the next. In this work, we adopt the perspective that the reasoning power of Transformers is fundamentally limited by a fixed maximum number of steps along any latent path of computation. To address this, we introduce Turbo Connection (TurboConn), a novel architecture that overcomes the fixed-depth constraint by routing multiple residual connections from the higher-layer hidden states of each token $t$ to the lower layers of token $t + 1$. Fine-tuning pre-trained LLMs with our method not only yields accuracy gains of 0.9% to over 10% on benchmarks like GSM8K, Parity, and multi-step arithmetic, but also demonstrates that the density of these backward connections is critical; our dense interaction significantly outperforms "sparse" alternatives that only pass a single hidden state or vector. Notably, TurboConn can be integrated into pre-trained LLMs to overcome task-specific plateaus: while a fine-tuned Qwen-3-1.7B achieves only 53.78% on Parity, adding our architectural modification enables the model to reach 100% accuracy, all without the necessity to retrain the full model from scratch or sophisticated curriculum learning. Our results provide strong empirical evidence that the depth of the computational path is a key factor in reasoning ability, also offering a new mechanism to enhance LLMs without significantly affecting generation latency.

## 1. Introduction

While Transformer-based Large Language Models (LLMs) have advanced significantly in recent years, they continue to

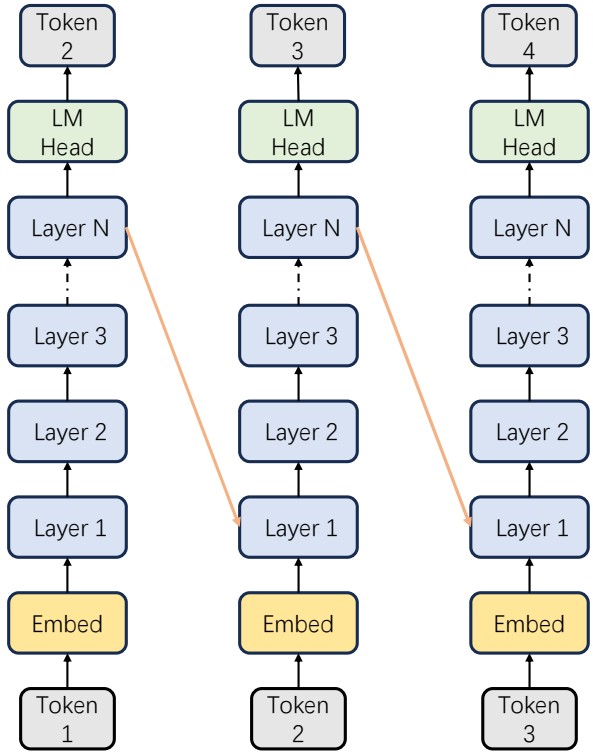

*Figure 1.* Modified Transformer architecture with downward connections (orange arrows) from higher to lower decoder layers.

exhibit shortcomings on tasks that demand complex reasoning. The popular Chain-of-Thought (CoT) framework (Wei et al., 2022) addresses this by allocating dynamic computation through intermediate steps. However, this approach places a considerable strain on computational resources and often requires specialized training data (DeepSeek-AI et al., 2025).

A growing area of research (Dehghani et al., 2019; Geiping et al., 2025; Fan et al., 2025a) focuses on performing additional computation in the depth dimension of a Transformer rather than along the token sequence dimension. This is achieved by recursively applying Transformer layers, which enables increased latent reasoning. In this approach, "thinking" occurs within the model's hidden states instead of being explicitly projected onto tokens, potentially allowing for more complex information processing. A key advan-

[1]UCLA. Correspondence to: Mohan Tang <tangmohanp@outlook.com>.

*Proceedings of the $43^{rd}$ International Conference on Machine Learning*, Seoul, South Korea. PMLR 306, 2026. Copyright 2026 by the author(s).

tage of this approach is that it allows training on unlabeled pre-training data (Geiping et al., 2025; Zeng et al., 2025). However, it still requires increased time and GPU costs during training, as well as longer inference times that scale proportionally with the number of recursion steps. These factors pose considerable scalability concerns for future development.

Is enhancing reasoning solely a matter of increasing the total amount of computation? We argue that the answer is no. In this work, we aim to increase the effective depth—defined as the maximum length of the computational path available to process information—with negligible impact on total floating-point operations. In standard Transformers, information flows strictly from lower to higher layers. As a result, the maximum length of the computational path is always bounded by a fixed number (proportional to the depth of the model). Because the model's depth is fixed regardless of input length, this architecture imposes limitations on the Transformer's computational power. Aligning with this perspective, Merrill & Sabharwal (2023) demonstrated that finite-depth Transformers are confined to the uniform $TC^0$ complexity class, a class of problems solvable by constant-depth circuits, suggesting they may lack the expressive power required for inherently sequential problems. Feng et al. (2023) also presents a viewpoint that CoT works by increasing the "effective depth" of Transformers as the outputs are repeatedly looped back to the input.

Overcoming this constraint, we introduce a novel architecture featuring connections from higher layers back to lower layers. Specifically, the hidden state from a higher layer of token $t$ is fed into a lower layer processing token $t+1$. This modification allows the effective reasoning depth to grow linearly with the sequence length, breaking the fixed-depth constraint of standard Transformers with only a marginal increase in total computation. We refer to our method as Turbo Connection (TurboConn). The $t \to t+1$ design is a deliberate choice that leverages the autoregressive nature of the Transformer decoder to avoid the circular dependencies that would arise from self-connections (e.g., $t \to t$), a point we discuss in detail in Section 3.2. The primary trade-off is the loss of full parallelism during training and the prefill stage of inference, as processing degenerates to group-sequential evaluations. Nevertheless, this has little impact on GPU memory usage or the speed of autoregressive generation.

A key advantage of TurboConn is that it requires no modifications to the standard loss function, making it compatible with existing pre-training objectives, though our experiments are focused on the fine-tuning stage due to resource constraints. For this work, we focus on evaluating its effectiveness by fine-tuning Llama 3 1B and 8B models (Grattafiori et al., 2024) and Qwen 3 1.7B model (Yang et al., 2025) on several reasoning-intensive datasets. We find

that our architecture consistently outperforms the standard Transformer architectures across these tasks.

To summarize our contributions:

1. We propose a novel method that increases the reasoning ability of large language models, without:
   - additional latency at inference-time autoregressive generation,
   - increased GPU memory consumption during training or inference, or
   - the need for human-annotated chain-of-thought data.
2. Across different models and tasks, TurboConn yields accuracy gains ranging from 0.9% to over 10% on reasoning benchmarks.
3. We show that TurboConn enables a qualitative shift in model behavior, including better *length generalization* on the Parity task and emergence of *discriminative filtering*.
4. Our results provide strong empirical evidence that the lack of depth in latent computation indeed limits the reasoning ability of LLMs. Having a fixed computational depth for all tokens is likely a necessary consequence of the standard parallel training paradigm for Transformers, where all tokens in a sequence are processed simultaneously. This finding could motivate more research to move beyond purely parallel designs and explore sequential or non-parallel training paradigms to unlock deeper reasoning.

## 2. Related Work

### 2.1. Chain-of-Thought

People observe that large language models benefit from generating more tokens to "think step by step" (Wei et al., 2022; Kojima et al., 2022). Theoretically, chain-of-thought has been shown to be able to greatly enhance LLMs in terms of computational power (Merrill & Sabharwal, 2024; Li et al., 2024). To train models to produce these reasoning chains, researchers have explored several paradigms. The most direct method is supervised fine-tuning on chain-of-thought data (Liu et al., 2025a; Yue et al., 2023; Ahmad et al., 2025). A more challenging but highly sought-after goal is to elicit CoT reasoning using only final outcomes for supervision (Zelikman et al., 2022; Zhu et al., 2023; Singh et al., 2024). The most prominent current approach in this domain is reinforcement learning from outcome-based rewards (OpenAI et al., 2024; DeepSeek-AI et al., 2025; Team et al., 2025; Liu et al., 2025b). Recent work has pushed these boundaries even further. For instance, some methods aim to train CoT behavior on unlabeled data using only the standard next-token prediction objective (Zelikman et al., 2024).

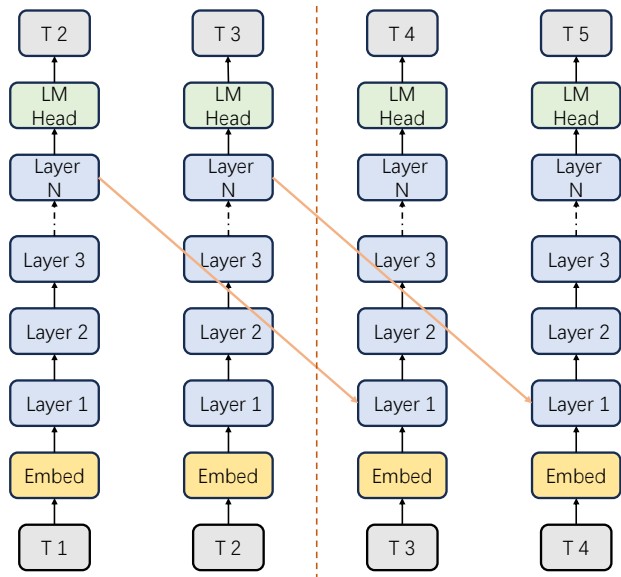

*Figure 2.* Grouping strategy for downward connections. Example shown for group of 2.

plications: assuming the standard conjecture that $L \neq P$, Transformers cannot solve problems such as linear equation solving or universal context-free grammar recognition. Feng et al. (2023) also shows that bounded-depth Transformers cannot solve certain basic mathematical tasks unless the model size grows super-polynomially with respect to the input length and explains why CoT works through the lens of increased "effective depth."

Empirical evidence also highlights the critical role of depth in how models perform multi-step reasoning. For instance, LLMs have been shown to resolve two-hop problems by handling the first hop in lower layers before processing the second hop in higher layers (Biran et al., 2024). Similarly, intensive training on multi-hop reasoning tasks leads to the gradual strengthening of intermediate representations in the middle layers, which are then used by subsequent layers to reach a final answer (Wang et al., 2025). Our work is distinguished from these approaches by isolating the effect of depth alone, without a significant increase in computation, to empirically show its importance.

Particularly relevant is the "back-patching" experiment by Biran et al. (2024), which showed that manually injecting a hidden state from a higher layer back into a lower layer can correct reasoning failures. Our work can be seen as operationalizing this insight: instead of using it as a post-hoc analysis or intervention that requires a second, full forward pass, we integrate this top-to-bottom connection directly into the model's architecture.

Among novel architectural designs, Fan et al. (2021) enables top-down information flow by replacing the Transformer's standard key-value memory with a single key-value set per token that encapsulates information across all layers. Similarly, Staircase Attention (Ju et al., 2022) introduces a recurrent architecture that applies a shared Transformer block iteratively over overlapping token groups, feeding hidden states from previous steps back into the core network as new tokens are processed. While both methods also provide feedback from higher to lower layers, our work is uniquely designed to augment existing, large-scale pre-trained LLMs. TurboConn successfully leverages expensive, pre-trained features of LLMs by using zero-initialized additive connections.

While the standard CoT framework relies on generation of discrete tokens, others have explored continuous Chain-of-Thought, where the discrete reasoning tokens are replaced with continuous representations derived from the model's hidden states (Hao et al., 2024). This method enables continuous reasoning by passing the top-layer hidden states back to the initial embedding layer on dedicated ⟨pause⟩ tokens inserted between problems and answers. However, training such continuous thinking is challenging, requiring extensive curriculum learning from existing CoT datasets.

### 2.2. Looped/Universal Transformers

Looped or universal Transformers apply the same set of Transformer layers recursively in the depth dimension before producing a final output. This architectural paradigm has shown advantages in reasoning tasks, as demonstrated in several experiments (Dehghani et al., 2019; Saunshi et al., 2025). Theoretically, looped Transformers have been proven capable of solving various logical problems of interest (Giannou et al., 2023; Fan et al., 2025b). More recently, variants of this approach have been successfully scaled up to the billion-parameter range and trained on large, general-purpose pre-training datasets (Geiping et al., 2025; Zeng et al., 2025).

### 2.3. On Relationship between Depth and Reasoning

Merrill & Sabharwal (2023) theoretically showed that finite-depth Transformers are confined to the uniform $\text{TC}^0$ complexity class, which is believed to have limited computational power. This theoretical constraint has practical im-

## 3. Method

### 3.1. Background and Notations

Consider the following formalization of the Transformer architecture. Let $\boldsymbol{h}_l^{(i)}$ be the hidden state of token $i$ at layer $l$ for $l = 0, \ldots, L$. Given input tokens $t_0, t_1, t_2, \ldots, t_k$, obtain their embeddings

$$\boldsymbol{h}_0^{(i)} = \boldsymbol{e}_i = \text{Embed}(t_i), \qquad i = 0, \ldots, k.$$

For $l = 1, \ldots, L$ and $i = 0, \ldots, k$,

$$h_l^{(i)} = \text{LayerBlock}_l\big(h_{l-1}^{(i)}; \, h_{l-1}^{(0)}, h_{l-1}^{(1)}, \ldots, h_{l-1}^{(i-1)}\big)$$

Here $\text{LayerBlock}(\cdot)$ encapsulates the self-attention, MLP, positional encodings, residuals, etc.

The final logits are

$$P^{(i)} = \text{lm\_head}\big(h_L^{(i)}\big), \qquad i = 0, \ldots, k.$$

### 3.2. Our Approach

We introduce a novel modification to the standard Transformer architecture by incorporating downward connections from higher layers to lower layers, as shown in Figure 1.

In contrast to our approach, an intuitive design might be to feed a token's higher layers back into its own lower layers ($t \rightarrow t$). However, this creates a circular dependency in the computational graph, a logical impossibility in a single forward pass. To resolve this loop, the model would need to re-run its layers multiple times for a single token, a process that dramatically increases the required computation at inference time. In contrast, TurboConn avoids this problem. The causal nature of the Transformer decoder means that token $t$ (past) can influence token $t + 1$ (future) without creating any circular dependency in the computational graph.

Formally, for $l = 1, \ldots, L$ and $i = 1, \ldots, k$, we modify the hidden state computation as such:

- When a connection from layer $s$ to layer $l$ exists:

$$h_l^{(i)} = \text{LayerBlock}_l\big(h_{l-1}^{(i)}; \, h_{l-1}^{(0)}, \ldots, h_{l-1}^{(i-1)}\big) \\ + \alpha \cdot \mathcal{D}\big(h_s^{(i-1)}\big)$$

- Otherwise (when no connection ($s \rightarrow l$) exists):

$$h_l^{(i)} = \text{LayerBlock}_l\big(h_{l-1}^{(i)}; \, h_{l-1}^{(0)}, \ldots, h_{l-1}^{(i-1)}\big)$$

$\mathcal{D}(\cdot)$ is a learnable linear projection specific to each connection. We can initialize $\mathcal{D}(\cdot)$ to be all zeros when finetuning, so that the model behaves the same as the original LLM at the start of training. The multiplier $\alpha$ scales the strength of downward connections. It does not affect the model's theoretical representational power, but encourages greater utilization of the connections during training. By default we set $\alpha$ to 100. The effect of different multiplier values is analyzed in Section 4.4. Note that there can be multiple connections, as shown in Figure 3.

Because $h_l^{(i)}$ depends only on tokens $0, \ldots, i$, the computation can proceed token-by-token from left to right. Turbo-Conn preserves the standard language modeling paradigm

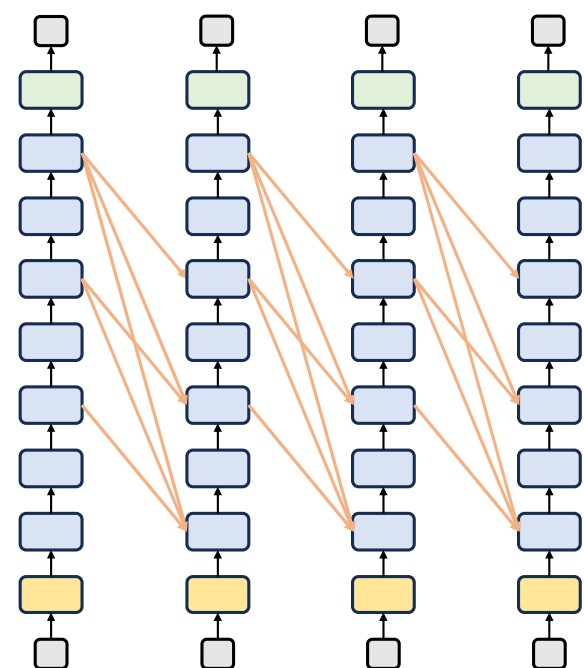

*Figure 3.* This diagram illustrates the dense connectivity pattern utilized in our experimental setup, depicted here for a group size of 1.

and can be trained using the conventional autoregressive loss:

$$\mathcal{L} = -\sum_{i=1}^{k} \log P(t_i \mid t_0, t_1, \ldots, t_{i-1})$$

#### 3.2.1. ANALYSIS OF DEPTH

As an example, consider the case that we have a connection from the highest layer to the lowest layer. In standard Transformers, the maximum computational path length is $L$ (corresponding to the number of layers). With TurboConn, this path length extends to $kL$, where $k$ is the sequence length. The maximum depth of reasoning now scales linearly with the number of input tokens. While this does not enable the model to solve all arbitrarily complex problems, the shift from a constant to a linear-depth computational model is a fundamental breakthrough compared to the standard Transformer.

### 3.3. Analysis of Computational Cost

The downward connections introduce sequential dependencies between tokens, breaking the parallelism typically available during training. This requires sequential processing of tokens during the forward pass, potentially increasing training latency. Similarly, the prefill stage during inference degenerates to sequential processing, introducing a

potential latency bottleneck, though this can be mitigated by evaluating tokens in larger groups. However, this sequential constraint does not impact the generation phase at inference time, as autoregressive generation inherently processes tokens one by one.

Regarding memory requirements, TurboConn introduces only marginal additional hidden states that need to be stored, resulting in minimal increase in GPU memory usage. The memory footprint remains comparable to standard Transformers.

### 3.4. Grouping

To mitigate the computational cost of sequential processing, we introduce a grouping mechanism that enables more efficient parallel computation. As shown in Figure 2, we group tokens together and send information back in groups rather than processing individual tokens sequentially. This approach processes tokens in groups of size $g$, where each group receives downward connections simultaneously. For layers $l = 1, 2, \ldots, L$ and token positions $i = g, g + 1, \ldots, k$, we modify the hidden state computation as follows: when a connection from layer $s$ to layer $l$ $(s \rightarrow l)$ exists:

$$\boldsymbol{h}_l^{(i)} = \text{LayerBlock}_l\big(\boldsymbol{h}_{l-1}^{(i)}; \boldsymbol{h}_{l-1}^{(0)}, \ldots, \boldsymbol{h}_{l-1}^{(i-1)}\big) + \alpha \cdot \mathcal{D}\big(\boldsymbol{h}_s^{(i-g)}\big)$$

In this way, $\boldsymbol{h}^{(i)}$ does not depend on hidden states at tokens $i - g + 1, i - g + 2, \ldots, i - 1$, so computations within each group can be performed in parallel. With group size $g$, the computational depth becomes $kL/g$ (assuming there is a connection from top to bottom), and the number of sequential steps reduces from $k$ to $k/g$. This reduces training latency while decreasing computational power. The group size can be chosen based on the specific use case requirements. Applications requiring deep sequential reasoning may benefit from smaller group sizes, while those prioritizing training efficiency may prefer larger groups.

### 3.5. Comparison with Universal Transformers

While both TurboConn and the Universal Transformer utilize recurrence to increase depth, the Universal Transformer is recurrent in depth (per token), whereas TurboConn is recurrent across the sequence (cross-token). To clarify the mechanical differences, we compare the two architectures under the assumption that both are configured to increase the effective reasoning depth by a factor of $D$.

#### 3.5.1. COMPUTATIONAL EFFICIENCY AND SCALING

To increase the effective reasoning depth by a factor of $D$, the Universal Transformer must perform $D$ recursive iterations for **every token**. Its training and inference time scale linearly with $D$. In contrast, TurboConn scales the depth by utilizing the sequence dimension. Although it breaks full token-level parallelism, it preserves intra-group and tensor-wise parallelism. Therefore, the training time increases by a factor significantly less than $D$.

#### 3.5.2. MEMORY FOOTPRINT

For the Universal Transformer to achieve a depth factor of $D$, the memory used must grow by a factor of $D$, as the number of states that must be stored for each recursion increases. Conversely, TurboConn introduces no significant change in memory cost.

#### 3.5.3. AUTOREGRESSIVE GENERATION AND INFERENCE

The difference is also pronounced during the autoregressive generation phase. For the Universal Transformer, each generated token requires $D$ passes through the recurrent layers, which increases the latency by a factor of $D$. TurboConn, however, involves little additional cost during generation. The backward connections are integrated into the single forward pass of the model, allowing for depth-scaling without the linear latency penalty of Universal Transformers.

## 4. Experiments

### 4.1. Reasoning Tasks

We evaluate the effectiveness of TurboConn on the following datasets. Each dataset consists of about 380K questions:

1. **Parity** (Fan et al., 2025a; Banino et al., 2021; Graves, 2017): Given a sequence of 0s and 1s, the task is to determine whether there is an odd number of 1s. We sample sequences ranging from 1 to 70 digits in length.

2. **Multi-step Arithmetic** (Srivastava et al., 2023): Randomly generated arithmetic expressions using digits 0-9, with the number of operands ranging from 1 to 30. The result is projected to modulo 10 for a more uniform distribution of answers.

3. **GSM8K** (Cobbe et al., 2021): A collection of grade school math problems. We use an enlarged dataset augmented by Deng et al. (2023). To evaluate performance without process supervision, we removed the chain-of-thought reasoning steps from the original dataset.

We train and evaluate on different splits of the same distribution. Further details on training data can be found in Appendix A.4.

## 4.2. Training Hyperparameters

We evaluate performance by fine-tuning pre-trained Llama 3 and Qwen 3 models, comparing a standard Transformer baseline against the Transformer augmented with Turbo-Conn under identical training settings. Training uses a batch size of 64, resulting in approximately 6,000 iterations per epoch for each dataset. We train for a maximum of 3 epochs, following standard fine-tuning practices to prevent overfitting.

Each model has a dense connection setup with 15 to 45 connections. This configuration empirically provides more stable training.

Due to resource constraints, we use LoRA on attention and feed-forward parameters. To ensure fair comparison, we set the rank to r = 120 for TurboConn and r = 140 for the baseline, maintaining comparable parameter counts. The downward connection projections $\mathcal{D}(\cdot)$ are also implemented as low-rank linear maps.

We employ a cosine scheduler (Loshchilov & Hutter, 2017), where the lr increases and decreases in periods of 1000 steps. In each period, it is warmed up from 0 to the highest value in 100 steps, and then gradually decrease to 0 again following a Cosine function with period 900.

Additional training details are provided in Appendix A.

## 4.3. Main Results

We evaluate our method on Llama 3.2 1B and Llama 3.1 8B models (Grattafiori et al., 2024) and Qwen 3 1.7B model. We use a group size of 4 and set the connection multiplier to 100. Results are presented in Table 2. TurboConn demonstrates significant improvements over standard Transformers across all model sizes and datasets. Notably, the 1B model with TurboConn outperforms the 8B model without connections on the Parity task, suggesting that improved information flow can be more beneficial than increased computational scale for certain reasoning tasks. Moreover, adding TurboConn to pretrained LLM enables Qwen-3-1.7B to achieve $100\%$ accuracy on Parity, where fine-tuning alone fails to resolve the underlying complexity and plateaus at $53.78\%$.

## 4.4. Effect of Group Sizes and Multiplier

In this section, we evaluate the effect of different group sizes $g$ on both model performance and computational efficiency. Furthermore, we find that the choice of group size dictates the optimal setting for the connection multiplier ($\alpha$). We conduct our analysis using the Llama 3.2 1B model across all three datasets.

To ensure consistent experimental setup, we remove a small number of overly long examples (ones exceeding 168 tokens using Llama 3.2 1B tokenizer) from GSM8K so that all experiments can be run with identical settings. Based on this threshold, only 44 examples were removed from the original dataset of 384,620. Each measurement is performed on a single 80G A100 GPU.

### 4.4.1. Training Latency Analysis

Table 1 presents computational efficiency analysis across different group sizes. The results demonstrate that larger group sizes lead to reduced training latency. Importantly, when we recurse for $k/g$ times, the training latency increases by less than a factor of $k/g$. This occurs because our sequential token processing preserves other forms of parallelism, including batch-level parallelism and vectorized operations within the attention and feed-forward computations. As shown in Table 1, on the Parity task, group size 4 requires 38.81 recursions yet increases training time by only 4.87×. Moreover, Parity with group size 16 achieves 10× deeper reasoning paths with just 1.36× training overhead–delivering enhanced reasoning capability at near-baseline computational cost. This reveals a favorable trade-off between reasoning depth and computational efficiency, suggesting that our method can provide substantial improvements in model reasoning capabilities with manageable increases in training time.

### 4.4.2. Performance Analysis

The results are shown in Table 1. In general, we can see that smaller group sizes result in stronger performance. One issue we observed is that when using a large multiplier ($\alpha = 100$) and a large group size, the performance can sometimes degrade compared to the baseline. This suggests that while the downward connections pass valuable, high-level information from previous tokens, the reduced computational depth of larger groups may be insufficient to process this information effectively. The influx of strong signals from higher layers can destabilize the training process. When using a high multiplier, the strong signals from downward connections may also cause the model to over-rely on the connections for information propagation between tokens at the expense of its standard attention mechanisms.

Conversely, setting the multiplier to a lower value ($\alpha = 1$) mitigates this issue, leading to consistent performance gains over the standard Transformer across all tested group sizes. On the other hand, this approach results in less significant improvements when group size is small, compared to using $\alpha = 100$. A smaller $\alpha$ value encourages the model to utilize the downward connections more subtly, resulting in steady, incremental advantages without destabilizing the training process. Therefore, the recommended approach is to pair a small group size with a large multiplier and a larger group size with a small multiplier.

*Table 1.* Training efficiency and performance analysis for Llama 3.2 1B across group sizes for multipliers $\alpha = 1$ and $\alpha = 100$. We report average sequence length, average number of recursions per training iteration, and relative training time per step compared to the baseline transformer across three reasoning tasks.

| Dataset (Avg. Seq. Length) | Method | # Recursions | Time/Step (seconds) (× Transformer) | Acc (%) ($\alpha = 1$) | Acc (%) ($\alpha = 100$) |
|---|---|---|---|---|---|
| Parity (154.69 tokens) | Baseline | 1.00 | 1.18 (1.00×) | 92.87 | – |
| | TurboConn (Group 4) | 38.81 | 5.75 (4.87×) | 98.94 | **100.00** |
| | TurboConn (Group 6) | 25.94 | 3.07 (2.60×) | 98.68 | **100.00** |
| | TurboConn (Group 8) | 19.84 | 2.66 (2.25×) | 98.83 | 51.42 |
| | TurboConn (Group 16) | 10.00 | 1.61 (1.36×) | 98.85 | 51.40 |
| Multi-step Arithmetic (125.75 tokens) | Baseline | 1.00 | 1.01 (1.00×) | 38.16 | – |
| | TurboConn (Group 4) | 31.81 | 4.46 (4.42×) | 40.13 | **42.66** |
| | TurboConn (Group 6) | 21.37 | 2.29 (2.27×) | 39.94 | 41.74 |
| | TurboConn (Group 8) | 16.13 | 1.82 (1.80×) | 39.79 | 41.48 |
| | TurboConn (Group 16) | 8.17 | 1.45 (1.44×) | 38.79 | 38.79 |
| GSM8K (No CoT) (101.92 tokens) | Baseline | 1.00 | 0.77 (1.00×) | 7.20 | – |
| | TurboConn (Group 4) | 25.86 | 3.22 (4.18×) | 7.67 | **8.32** |
| | TurboConn (Group 6) | 17.40 | 1.73 (2.25×) | 7.87 | 7.95 |
| | TurboConn (Group 8) | 13.18 | 1.39 (1.81×) | 7.55 | 8.17 |
| | TurboConn (Group 16) | 6.84 | 1.38 (1.80×) | 7.41 | 6.94 |

## 4.5. Comparison to Single "Soft-Token" Feedback

Chain-of-thought (CoT) can also be conceptualized as a mechanism for passing information from higher to lower layers, by feeding back the information of a single discrete token into the input embedding of the next step. Similarly, methods like Coconut (Hao et al., 2024) utilize a continuous CoT approach, passing the highest hidden state back to the lowest layer, effectively passing information on the distribution of the next token. In contrast, our method is more "dense," passing hidden states from multiple layers back to multiple layers simultaneously. To determine if this architectural density is providing extra reasoning power, we compare our approach against a single "soft-token" feedback baseline.

We implement a feedback mechanism that passes only a single vector of information from the top of the model back to the input. Specifically, the input to the first layer for token $i$, denoted as $h_0^{(i)}$, is modified as follows:

$$h_0^{(i)} = (1 - \lambda)e_i + \lambda \sum_{j \in \mathcal{V}} P(x_i = j | x_{<i}) \text{Emb}(j)$$

where $e_i$ is the original embedding of token $i$, Emb is the embedding matrix, and $\lambda$ is a scaling factor (set to 0.1) representing the strength of the feedback. In this setup, $\lambda = 1$ would imply the prediction of the previous token

completely overwrites the current input.

### 4.5.1. RESULTS AND ANALYSIS.

As shown in Table 3, the single soft-token approach is insufficient to capture the reasoning gains of our dense architecture. These results suggest that multi-layer downward connections offer a more effective mechanism for unlocking the reasoning power of depth-scaling than the sparse feedback of a single vector or distribution.

## 4.6. Length Generalization Experiment

We hypothesize that models with TurboConn learn fundamentally different problem representations compared to the standard Transformer. This is particularly evident in the Parity task, where our method achieves perfect accuracy. To further verify this claim, we conducted a length generalization experiment using the Llama 3.1 8B model. Turbo-Conn (with a group size of 1), "soft-token" method, and the baseline Transformer were all trained exclusively on parity sequences with a maximum length of 10. We then evaluated their performance on much longer sequences.

The results are shown in Figure 4. While all models achieved near-perfect accuracy on the training data, their generalization capabilities diverged significantly. As we can

*Table 2.* Performance comparison across model sizes and datasets, comparing standard Transformers baseline against Transformers augmented with TurboConn.

| Model | Dataset | Method | Acc (%) |
|---|---|---|---|
| Llama 3.2 1B | GSM8K (No CoT) | Baseline | 7.20 |
| | | TurboConn | **8.32** |
| | Multi-step-arithmetic | Baseline | 38.16 |
| | | TurboConn | **42.66** |
| | Parity | Baseline | 92.87 |
| | | TurboConn | **100.0** |
| Llama 3.1 8B | GSM8K (No CoT) | Baseline | 23.92 |
| | | TurboConn | **24.82** |
| | Multi-step-arithmetic | Baseline | 48.32 |
| | | TurboConn | **51.86** |
| | Parity | Baseline | 89.40 |
| | | TurboConn | **100.0** |
| Qwen 3 1.7B | GSM8K (No CoT) | Baseline | 15.90 |
| | | TurboConn | **20.31** |
| | Multi-step-arithmetic | Baseline | 36.10 |
| | | TurboConn | **45.81** |
| | Parity | Baseline | 53.78 |
| | | TurboConn | **100.0** |

*Table 3.* Comparison of our dense feedback method against standard Transformers baseline and the single soft-token feedback using Llama 3.2 1B.

| Model | Dataset | Method | Acc (%) |
|---|---|---|---|
| Llama 3.2 1B | GSM8K (No CoT) | Baseline | 7.20 |
| | | Soft-token | 7.07 |
| | | **TurboConn** | **8.32** |
| | Multi-step Arithmetic | Baseline | 38.16 |
| | | Soft-token | 38.00 |
| | | **TurboConn** | **42.66** |
| | Parity | Baseline | 92.87 |
| | | Soft-token | 96.51 |
| | | **TurboConn** | **100.0** |

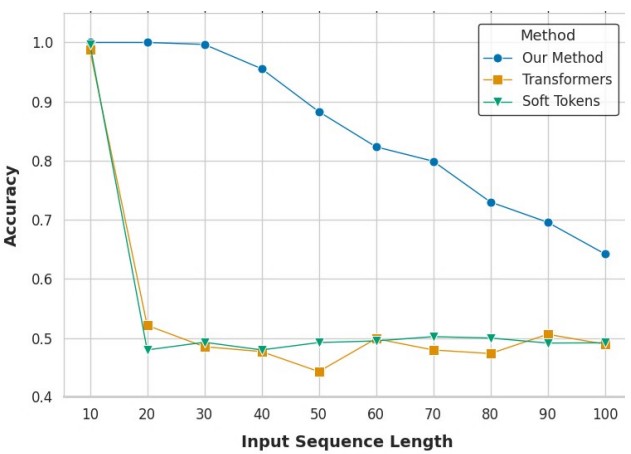

*Figure 4.* Length Generalization Performance. We evaluate a Llama 3.1 8B model, trained on Parity with up to 10-digit sequences, on its ability to generalize to longer input sequences.

see from the plot, the LLM with TurboConn shows much better length generalization ability than Transformer and soft-token method. In particular, our method maintains perfect accuracy on sequences up to length 30. This suggests that TurboConn allows the model to learn a more robust and generalizable algorithm for the task, whereas the standard Transformer appears to overfit to the training distribution, leveraging its vast parameter count to solve the problem.

### 4.7. Emergence of Discriminative Filtering

Beyond improvements in top-1 accuracy, we observe that TurboConn results in a fundamental change to the output distribution of the models, and enables the emergence of *discriminative filtering*. We define this as the model's ability to confidently eliminate mathematically incorrect candidates from the output probability distribution.

To quantify this effect, we measure the number of "eliminated choices," defined as the count of digits $\{0, \ldots, 9\}$ assigned a probability below a threshold of $\tau = 0.001$ by the models after finetuning. As shown in Table 4, on the Multi-step Arithmetic task, Qwen-1.7B model augmented with TurboConn eliminates an average of **5.259** choices per token, more than doubling the **2.516** choices eliminated by

the standard Transformer baseline.

A particularly striking result is that while increasing the model scale allows standard Transformers to match our accuracy, their discriminative power remains significantly lower. Qwen 3 1.7B model with TurboConn eliminates more choices (5.259) than the 8B baseline (4.500), despite achieving nearly identical accuracy (45.81% vs 45.61%). This signals a fundamental gap in this reasoning dimension that cannot be easily filled by parameter scaling alone, suggesting that the depth afforded by our method enables a more efficient internal verification process.

### 4.8. Synergy with Chain-of-Thought Reasoning

While TurboConn enhances the internal reasoning capabilities of models without explicit intermediate tokens, it is also compatible with standard Chain-of-Thought prompting.

*Table 4.* Comparison of accuracy and discriminative filtering on Multi-step Arithmetic after finetuning. "Eliminated Choices" represents the mean number of digit classes with probability $P < 0.001$. The result for TurboConn is reported using a group size of 4.

| Model | Method | Acc (%) | Eliminated Choices |
|-------|--------|---------|--------------------|
| Qwen3-1.7B | Baseline | 36.10 | 2.516 |
| **Qwen3-1.7B** | **TurboConn** | **45.81** | **5.259** |
| Qwen3-4B | Baseline | 43.68 | 4.609 |
| Qwen3-8B | Baseline | 45.61 | 4.500 |

One way to combine those methods is to augment the explicit reasoning of CoT with the latent reasoning ability of backward connections. We hypothesize that TurboConn can act as an error-correction mechanism for explicit reasoning chains, allowing the model to more accurately follow the "program" laid out in the CoT.

To test this synergy, we utilize the NuminaMath-CoT dataset (LI et al., 2024) and augmented GSM8K (Deng et al., 2023) dataset to generate CoT trajectories. We first fine-tune a Llama 3.2-1B model to produce reasoning chains for mathematical problems. We then compare models with Turbo-Conn against the standard Transformers baseline by fine-tuning both to predict the final numerical outcome based on the same generated CoTs. This setup ensures that both models process the same explicit reasoning steps, isolating the effect of latent computation on final answer derivation. More details on this experiment are provided in Appendix A.7.

As shown in Table 5, TurboConn improves the model's ability to reach the correct conclusion from a given reasoning chain. These results suggest that TurboConn can effectively use latent computation to verify intermediate steps and correct potential errors within the explicit reasoning process, leading to more robust mathematical performance.

We note that the performance gains on GSM8K are less pronounced compared to those on NuminaMath. Since the training data for GSM8K is augmented using GPT-4, there is a significant distribution shift between the augmented training data and the original test/validation splits, potentially reducing the utility of the learned representations. We believe that our method would benefit more from access to sufficient higher-quality data that closely matches the target distribution.

### 4.8.1. GENERALIZATION TO COMPETITION MATHEMATICS

To further evaluate the robustness of our method, we test the checkpoints trained on the NuminaMath dataset on recent competition mathematics problems. This evaluation set is an aggregate of problems from the 2023 AMC and 2024–

2025 AIME competitions (Mathematical Association of America, 2023; 2024; 2025). We employ the same "correct existing CoT" setting described above, utilizing the Llama 3.2-1B model finetuned on NuminaMath to generate the initial CoT trajectories, and maintaining identical inference hyperparameters.

As shown in Table 5, on this combined competition dataset, TurboConn demonstrates a significant relative performance gain (4.45% vs. 2.00%). These results indicate that even when a 1B parameter model struggles to generate high-quality explicit reasoning chains for highly complex problems, TurboConn's latent reasoning mechanism is still capable of extracting value from imperfect trajectories to correct errors and improve final performance.

*Table 5.* Performance comparison using generated Chain-of-Thought trajectories. All results use Llama 3.2-1B; TurboConn uses a group size of 8.

| Task | Method | Acc (%) |
|------|--------|---------|
| NuminaMath-CoT | Baseline + CoT | 30.96 |
| | **TurboConn + CoT** | **33.98** |
| GSM8K | Baseline + CoT | 36.45 |
| | **TurboConn + CoT** | **36.96** |
| AMC & AIME (Transferred) | Baseline + CoT | 2.00 |
| | **TurboConn + CoT** | **4.45** |

## 5. Conclusions

In this work, we introduce TurboConn, a novel method that creates residual connections from higher to lower layers between sequential tokens, allowing the model's effective reasoning depth to scale linearly with sequence length. Our results demonstrated that this architectural change yields significant performance gains on logical reasoning tasks, with only a manageable increase in computational cost. This confirms empirically that extending effective depth alone is a viable strategy for overcoming the inherent reasoning bottlenecks of fixed-depth Transformers. It would be valuable to explore the potential of this architecture further. Future work could focus on integrating this architecture into the pre-training phase, which may foster the development of more foundational, general-purpose reasoning skills.

### Acknowledgment

This work was conducted during the authors' studies at the University of California, Los Angeles. The authors acknowledge Peng's Language Understanding & Synthesis Lab at UCLA for providing the computational resources and infrastructure necessary for the experiments in this study.

## Impact Statement

This paper presents work whose goal is to advance the field of machine learning. There are many potential societal consequences of our work, none of which we feel must be specifically highlighted here.

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

# A. More Training Details

## A.1. Pseudocode Implementation

Algorithm 1 provides the complete pseudocode for our primary training procedure.

---

**Algorithm 1** Forward Pass with Cross-Layer Latent Bridges

---

1: **Input:** Token sequence $\mathbf{X} \in \mathbb{N}^{B \times L}$, group size $G$, bridge set $\mathcal{B} = \{(src, dest)\}$, multiplier $\alpha$
2: **Initialize:** KV cache $\mathcal{K} \leftarrow \emptyset$, logits list $\mathcal{Y} \leftarrow [\,]$
3: **Initialize:** State buffers $\mathbf{S}_{s,d} \leftarrow$ NULL for all $(s, d) \in \mathcal{B}$
4: **for** $t = 0, G, 2G, \ldots, L - G$ **do**
5:                                                        *// Extract current group of tokens*
6:     $\mathbf{X}_g \leftarrow \mathbf{X}_{:,\, t:t+G}$
7:     $\mathbf{H}^{(0)} \leftarrow$ EmbedTokens$(\mathbf{X}_g)$
8:     **for** $i = 0$ **to** $N - 1$ **do**
9:                                            *// 1. Standard Transformer Computation*
10:         $\mathbf{H}^{(i+1)}, \mathcal{K} \leftarrow$ TransformerLayer$_i(\mathbf{H}^{(i)}, \mathcal{K})$
11:                                   *// 2. Injection Phase: Inject latent state from previous group*
12:         **for** $(s, d) \in \mathcal{B}$ **do**
13:             **if** $i = d$ **and** $\mathbf{S}_{s,d} \neq$ NULL **then**
14:                 $\mathbf{H}^{(i+1)} \leftarrow \mathbf{H}^{(i+1)} + \alpha \cdot$ UpProj$_{s,d}(\mathbf{S}_{s,d})$
15:             **end if**
16:         **end for**
17:                                   *// 3. Extraction Phase: Save state for next group*
18:         **for** $(s, d) \in \mathcal{B}$ **do**
19:             **if** $i = s$ **then**
20:                 $\mathbf{S}_{s,d} \leftarrow$ DownProj$_{s,d}(\mathbf{H}^{(i+1)})$
21:             **end if**
22:         **end for**
23:     **end for**
24:     Append LMHead$(\mathbf{H}^{(N)})$ to $\mathcal{Y}$
25: **end for**
26: **Output:** Concat$(\mathcal{Y}, \mathrm{dim} = 1)$                      *// Shape: $[B, L, V]$*

---

## A.2. Hyperparameters

We set the learning rate for Llama 3 models to $1.92 \times 10^{-5}$. This was determined by scaling a base learning rate of $3 \times 10^{-7}$ proportionally with our batch size of 64. For Qwen 3 models, the learning rate is set to $5.12 \times 10^{-6}$, which is $8 \times 10^{-8} \times 64$. We set the decoding temperature to 1.0.

## A.3. Model Implementation

We implement TurboConn using a cache. During the forward pass, as the model processes each token, the cache stores the hidden states from the specified higher layers. When processing subsequent tokens, these cached states are retrieved and added to the corresponding lower-layer inputs.

The sequential nature of TurboConn necessitates the use of a key-value (KV) cache during the training forward pass, a mechanism typically reserved only for autoregressive inference. Because tokens (or groups of tokens) are processed sequentially, the attention keys and values from all previous tokens must be cached to be available for the current token's self-attention computation. To implement this efficiently, we store past key and value states in a list of tensors rather than concatenating them. This approach avoids the memory duplication that can occur when new tensors are created through repeated concatenation.

## A.4. Example Training Data

Here we offer some examples for the format of data used for training and evaluation.

### A.4.1. PARITY

**Prompt:**
Question: Output the parity of this sequence.
Input: 1 0 0 1 0 1
Answer:

**Completion:**
1

### A.4.2. MULTI-STEP ARITHMETIC

**Prompt:**
Question: Evaluate this expression modulo 10.
Input: (-(3 + -(1)) * 4 * (8 + 2 + 9)) =
Answer:

**Completion:**
8

### A.4.3. GSM8K

**Prompt:**
Question: Answer the math question.
Input: John cuts his grass to 2 inches. It grows .5 inches per month. When it gets to 4 inches he cuts it back down to 2 inches. It cost $100 to get his grass cut. How much does he pay per year?
Answer:

**Completion:**
300

## A.5. Connection Configurations

The specific wiring configurations were derived from a simple heuristic. Our principle was to ensure broad layer coverage while maintaining a low computational overhead:

$$\mathcal{C} = \{(s,l) \mid s - l > k, \text{ where } s, l \in [L_{min}, L_{max}] \text{ sampled at interval } m\}$$

We did not conduct an extensive search for "optimal" wiring. The only decision based on experiment is whether to leave the bottom-most layers unchanged. They are generally understood to encode basic features rather than the high-level logic targeted by our recurrence. When training appears unstable, we increase $L_{min}$ from 0 to a value that feels right. This turns out to be sufficient for our method to work.

The other hyperparameters of the connection formula were arbitrarily chosen at the beginning of our experiments. Our primary constraints were to ensure that the total number of trainable parameters did not exceed the standard baseline and that the connections covered the layers relatively evenly. Then these configurations were kept constant throughout all subsequent experiments. Because we conducted one or zero trials per model to determine these hyperparameters, we believe the method is robust to the specific wiring setup.

The specific wiring configurations for different models are listed below. The downward connections are defined from a source layer to a destination layer using the notation: 'source -> destination'.

### A.5.1. LLAMA 3.2 1B (16 TOTAL LAYERS)

A total of 15 connections were used, as specified below:

```
6 -> 0      8 -> 0      8 -> 2      10 -> 0     10 -> 2
```

```
10 -> 4     12 -> 0     12 -> 2     12 -> 4     12 -> 6
14 -> 0     14 -> 2     14 -> 4     14 -> 6     14 -> 8
```

### A.5.2. LLAMA 3.1 8B (32 TOTAL LAYERS)

A total of 45 connections were used, as specified below:

```
14 -> 7     16 -> 7     16 -> 9     18 -> 7     18 -> 9
18 -> 11    20 -> 7     20 -> 9     20 -> 11    20 -> 13
22 -> 7     22 -> 9     22 -> 11    22 -> 13    22 -> 15
24 -> 7     24 -> 9     24 -> 11    24 -> 13    24 -> 15
24 -> 17    26 -> 7     26 -> 9     26 -> 11    26 -> 13
26 -> 15    26 -> 17    26 -> 19    28 -> 7     28 -> 9
28 -> 11    28 -> 13    28 -> 15    28 -> 17    28 -> 19
28 -> 21    30 -> 7     30 -> 9     30 -> 11    30 -> 13
30 -> 15    30 -> 17    30 -> 19    30 -> 21    30 -> 23
```

### A.5.3. QWEN 3 1.7B (28 TOTAL LAYERS)

A total of 21 connections were used, as specified below:

```
12 -> 4     15 -> 4     15 -> 7     18 -> 4     18 -> 7
18 -> 10    21 -> 4     21 -> 7     21 -> 10    21 -> 13
24 -> 4     24 -> 7     24 -> 10    24 -> 13    24 -> 16
27 -> 4     27 -> 7     27 -> 10    27 -> 13    27 -> 16
27 -> 19
```

## A.6. LoRA Setting

For both 1B and 8B models, we use $r = 120$ for TurboConn and $r = 140$ for the original model. In this section, we show that with our setup, the number of trainable parameters available to TurboConn is less than or equal to that of the standard Transformer baseline.

In both setups, LoRA was applied to several primary parameters within each Transformer block.

There are the following key dimensions:

- $d_{hidden}$: The hidden size of the model.
- $d_{kv}$: The dimension of the key/value heads.
- $d_{inter}$: The intermediate size of the MLP blocks.

*Table 6.* Dimensions of the layers adapted with LoRA.

| Module | $d_{in}$ | $d_{out}$ |
|---|---|---|
| *Attention Block* | | |
| q_proj | $d_{hidden}$ | $d_{hidden}$ |
| k_proj | $d_{hidden}$ | $d_{kv}$ |
| v_proj | $d_{hidden}$ | $d_{kv}$ |
| o_proj | $d_{hidden}$ | $d_{hidden}$ |
| *MLP Block* | | |
| gate_proj | $d_{hidden}$ | $d_{inter}$ |
| up_proj | $d_{hidden}$ | $d_{inter}$ |
| down_proj | $d_{inter}$ | $d_{hidden}$ |

It is noted that all of these targeted modules in the base architecture are configured without bias terms.

The matrices for those parameters are unbiased. Given a LoRA rank $r$, the total number of trainable parameters per Transformer block ($P_{block}$) is calculated as follows:

$$P_{block} = \underbrace{4rd_{hidden}}_{\text{for q\_proj, o\_proj}} + \underbrace{2r(d_{hidden} + d_{kv})}_{\text{for k\_proj, v\_proj}} + \underbrace{3r(d_{hidden} + d_{inter})}_{\text{for gate, up, down\_proj}}$$

The total number of trainable parameters is then derived from this block-level calculation. For our method, we add the parameters from the $N_{conn}$ downward connections. Each connection is a LoRA-adapted linear layer ($d_{hidden} \rightarrow d_{hidden}$) with an added bias. The parameter calculation is:

$$P_{connection} = \underbrace{2rd_{hidden}}_{\text{LoRA matrices}} + \underbrace{r}_{\text{bias}} + \underbrace{d_{hidden}}_{\text{bias}}$$

### A.6.1. LLAMA 3.2 1B

For the Llama 3.2 1B model, the key dimensions are:

- $d_{hidden} = 2048$
- $d_{kv} = 512$
- $d_{inter} = 8192$

The number of trainable LoRA parameters per Transformer block ($P_{block}$), given a rank $r$, is calculated as:

$$P_{block} = 4r(2048) + 2r(2048 + 512) + 3r(2048 + 8192)$$
$$= 44032r$$

Baseline Model ($r = 140$):

$$P_{\text{total}} = 16 \times (44032 \times 140)$$
$$= \mathbf{98{,}631{,}680}$$

Our Method ($r = 120$):

$$P_{\text{conn}} = N_{\text{conn}} \times (2rd_{\text{hidden}} + r + d_{\text{hidden}})$$
$$= 15 \times (2 \cdot 120 \cdot 2048 + 120 + 2048)$$
$$= 7{,}405{,}320$$

The total number of trainable parameters is therefore:

$$P_{\text{ours}} = (16 \times P_{\text{block}}) + P_{\text{conn}}$$
$$= (16 \times 44032 \times 120) + 7{,}405{,}320$$
$$= \mathbf{91{,}946{,}760}$$

### A.6.2. LLAMA 3.1 8B

For the Llama 3.1 8B model, the key dimensions are:

- $d_{hidden} = 4096$
- $d_{kv} = 1024$
- $d_{inter} = 14336$

The number of trainable LoRA parameters per Transformer block ($P_{block}$), given a rank $r$, is calculated as:

$$P_{block} = 4r(4096) + 2r(4096 + 1024) + 3r(4096 + 14336)$$
$$= 81920r$$

Baseline Model ($r = 140$):

$$P_{\text{total}} = 32 \times (81920 \times 140)$$
$$= \mathbf{367{,}001{,}600}$$

Our Method ($r = 120$):

$$P_{conn} = N_{conn} \times (2rd_{hidden} + r + d_{hidden})$$
$$= 45 \times (2 \cdot 120 \cdot 4096 + 120 + 4096)$$
$$= 44,426,520$$

The total number of trainable parameters is therefore:

$$P_{ours} = (32 \times P_{block}) + P_{conn}$$
$$= (32 \times 81920 \times 120) + 44,426,520$$
$$= \mathbf{358,999,320}$$

### A.6.3. QWEN 3 1.7B

For the Qwen 3 1.7B model, the key dimensions are:

- $d_{hidden} = 2048$
- $d_{kv} = 1024$
- $d_{inter} = 6144$

The number of trainable LoRA parameters per Transformer block ($P_{block}$), given a rank $r$, is calculated as:

$$P_{block} = 4r(2048) + 2r(2048 + 1024) + 3r(2048 + 6144)$$
$$= 38912r$$

Baseline Model ($r = 140$):

$$P_{total} = 28 \times (38912 \times 140)$$
$$= \mathbf{152,535,040}$$

Our Method ($r = 120$):

$$P_{conn} = N_{conn} \times (2rd_{hidden} + r + d_{hidden})$$
$$= 21 \times (2 \cdot 120 \cdot 2048 + 120 + 2048)$$
$$= 10,367,448$$

The total number of trainable parameters is therefore:

$$P_{ours} = (28 \times P_{block}) + P_{conn}$$
$$= (28 \times 38912 \times 120) + 10,367,448$$
$$= \mathbf{141,111,768}$$

### A.7. More Training Details for TurboConn + CoT Experiments

Since the original NuminaMath-CoT dataset contains a test set of only 100 problems, we perform a custom re-split of the combined training and test data to ensure more statistically stable evaluations. We randomly partitioned the dataset into $384,000$ training, $1,000$ validation, and $1,000$ testing examples. The fixed CoT was generated by a Llama 3.2-1B model fine-tuned for 3 epochs on the original training sets, using a learning rate of $6.4 \times 10^{-4}$.

For GSM8K, we maintain the same experimental setup as for NuminaMath-CoT. We use the augmented GSM8K dataset described in Deng et al. (2023). However, unlike NuminaMath-CoT, GSM8K exhibits more significant overfitting to the training data; therefore, we use only 1/6 of the data to train the CoT generation model for a single epoch.

The training setup for the final prediction models remains identical to all previous experiments: we set the learning rate for Llama 3.2-1B models to $1.92 \times 10^{-5}$, use a batch size of 64, and train for 3 epochs using all training data.

In this particular experiment, we utilized a temperature of $0.01$ for evaluation to sharpen the probability distribution. This encourages the model to be more 'certain' in its predictions, often yielding higher expected accuracy by amplifying the likelihood of the dominant choice.

