# OpenReview forum: "Turbo Connection: Reasoning as Information Flow from Higher to Lower Layers"
_ICML.cc/2026/Conference — ICML 2026 regular_

### Official Review · Reviewer_WzN3 · 2026-03-02

**Soundness:** 3
**Presentation:** 3
**Significance:** 4
**Originality:** 3
**Overall Recommendation:** 5
**Confidence:** 3

**Summary:**

The paper introduces Turbo Connection, an architectural modification for Transformers that routes multiple residual connections from the higher-layer hidden states of a token t to the lower layers of the subsequent token t+1. While the wiring of the connection is fixed, the connection itself is a learnable projection. To manage the training latency caused by sequential token processing, the method incorporates a grouping mechanism that batches tokens into sizes of g and applies the downward connections simultaneously across each group. The architecture is evaluated by fine-tuning pre-trained models of different sizes (llama, Qwen). The modified models and the baselines are tested on a Parity, Multi-step Arithmetic, and GSM8K datasets. The experiments yield accuracy increases ranging from 0.9% to over 10% across these benchmarks, alongside measurable changes in length generalization capabilities on the Parity task.

**Compliance With Llm Reviewing Policy:**

Affirmed.

**Final Justification:**

The authors cleared all of my concerns. I am raising my score to an accept.

**Key Questions For Authors:**

- Q5: Regarding the impressive length generalization results, is there a specific training sequence length greater than 10 at which the baseline Transformer also begins to generalize to longer inputs?
- Q6: While the setup for connections without grouping is introduced, the main results report on a group size of 4. It would be quite interesting to explicitly see the performance difference between a per-token connection and the group connections.
- Q7: Given the long gradient paths introduced by connections through tokens, were there any observed issues with exploding or vanishing gradients during training?
- Q8: Did the authors test the effectiveness of this approach on (smaller) Transformers trained entirely from scratch on reasoning problems like Parity, rather than just fine-tuning?

Please also address the highlighted questions in the Weaknesses. I'm open to raising my score if the related work has been compared to in more detail.

**Limitations:**

yes

**Strengths And Weaknesses:**

Strength:

- The authors explore and experimentally validate a simple but elegant and effective idea. The architectural choice to route connections from token t to token t+1 bypasses the circular dependencies of the Universal Transformer. An ablation on Coconut-inspired soft tokens also shows the advantage over this approach.
- Very well motivated and generally well written. The paper is easy to follow.
- The length generalization results are super interesting and a definite highlight of the paper. They show a qualitative shift in the model's underlying capabilities and suggest a fundamentally different "reasoning strategy".
- An advantage for adoption is the method's compatibility with existing pre-trained LLMs. It successfully unlocks deeper reasoning capabilities without the heavy computational burden of retraining the full model from scratch.
- While the architecture inherently sacrifices full parallelization during training, the authors are up-front about this and transparently measure the training times. The introduction of the grouping mechanism is an effective and highly practical trade-off. The approach does have minimal to no additional cost in the inference stage.

Weaknesses

 - Figure 1 is overly abstract and lacks conceptual or structural information. The illustration of literal engine blocks is confusing and feels unprofessional in this context. It is highly recommended to replace this with a visually appealing diagram that provides an actual architectural overview of the method. Figures 2 and 3 are much better suited, though slightly misleading, as they depict only a single downward connection per pair of layers.
 - The authors need to compare their approach to related work on Recurrent-Transformer hybrids in more detail. The related work on Looped/Universal transformers is quite shallow and does not address the architectural differences and similarities. Additionally the authors missed some important work in this area:
      - Fan, Angela, et al. "Addressing some limitations of transformers with feedback memory." arXiv preprint arXiv:2002.09402 (2020).
       - Ju, Da, et al. "Staircase attention for recurrent processing of sequences." Advances in neural information processing systems 35 (2022): 13203-13213.
An experimental comparison would be preferable, although a deeper comparison on architectural differences would suffice.
 - In Section 3.2, the variable s is not explicitly defined, although it can be inferred from context. Furthermore, Appendix A.4.1 reveals a specific pattern to how connections are wired. **Q1:** How did the authors come up with this configurations? How much effort is required to find the optimal wiring for a given model? If there is a systematic pattern (e.g., every two layers between specific bounds), this should be formulated as a concise equation in the main text. **Q2:** Additionally, are there ablation studies on the "density of wires"?
 - While the authors compare TurboConn to a single "soft-token" feedback method (Coconut-inspired), they do not include comparisons against looped or universal transformers. Since looped architectures are prominently discussed in the related work as a primary method for increasing reasoning depth, their absence as a baseline makes it difficult to fully assess the performance-vs-efficiency trade-offs of the proposed method. At least a dedicated discussion should be included on the computational expense between the approaches.
 - The chosen benchmarks are somewhat dated given the rapid pace of LLM research. Including newer CoT benchmarks and broadening the scope of the evaluation to more general reasoning would be super interesting.
 - On page 6, the authors mention removing "a small number of overly long examples" from GSM8K to ensure a consistent experimental setup. **Q3:** Please specify exactly how many examples were removed and define the threshold for what constitutes "overly long" and include this in the paper.
 - The experiments comparing the method against standard Chain-of-Thought are somewhat weak. The accuracy advantages appear slight, and the experimental setup lacks clarity and could use some polishing. **Q4:** Where does the "distribution shift" come from and why does this "distribution shift affects the TurboConn approach more than the baseline llama?

---

> ### Author Rebuttal · Authors · 2026-03-31
>
> We thank Reviewer WzN3 for their thoughtful feedback and constructive suggestions. We address your feedback in detail below:
>
> ---
>
> ### **Re: Figure 1 is overly abstract**
> To improve clarity, we have removed Figure 1 and promoted the original Figures 2 and 3 to Figures 1 and 2, respectively. We have also added a new Figure 3 to demonstrate the dense connection pattern (anonymous URL: https://osf.io/xgrh2/overview?view_only=277783a27533493bac47c7d1a69c19c6).
>
> ---
>
> ### **Re: Comparison to existing Recurrent-Transformer hybrids**
> We thank the reviewer for pointing out these relevant works that were missed. While both methods also provide feedback from higher to lower layers, TurboConn is uniquely designed to augment existing, large-scale pre-trained LLMs. TurboConn leverages expensive, pre-existing features by using zero-initialized additive connections.
>
> **Comparing against Universal Transformer**
>
> Please refer to our response to Reviewer 4mDc for a detailed discussion on this point.
>
> ---
>
> ### **Re: Q1&Q2: Choice of connection locations**
>
> **Q1**
>
> We will clarify that $s \to l$ denotes a bridge from the output of layer with index $s$ to the input of layer with index $l$.
>
> The specific wiring configurations were derived from a simple heuristic. Our principle was to ensure broad layer coverage while maintaining a low computational overhead:
> $$\mathcal{C} = \\{ (s, l) \mid s - l > k, \text{ where } s, l \in [L_{min}, L_{max}] \text{ sampled at interval } m \\}$$
>
> **Q2**
>
> The comparison with the **soft-token** approach functions as an ablation for wiring density. While the soft-token method represents a more restricted communication channel, TurboConn’s multi-bridge setup provides a higher density of information flow across the sequence.
>
> ---
>
> ### **Re: The chosen benchmarks are somewhat dated**
>
> We have launched an experiment on generalization to more recent math benchmarks. Due to the computational cost, we could not complete this within the rebuttal periods, but commit to including the results in the final version.
> Given that current datasets consistently demonstrate that increased depth yields performance gains under identical training setups, we expect this principle to hold on more recent datasets as well.
>
> ---
>
> ### **Re: Filter method for GSM8K**
>
> We define "overly long" as exceeding 168 tokens using Llama 3.2 1B tokenizer. Based on this threshold, only 44 examples were removed from the original dataset of 384,620.
>
> ---
>
> ### **Re: Explain “distribution shift”**
>
> The distribution shift stems from using synthetic training data for GSM8K while evaluating on the original test set. This shift can cause models to learn "pseudo-rules" that are valid only within the training distribution but are not mathematically general. While this affects both models, it can diminish the observable gains of TurboConn.
>
> Regarding Section 4.8, we clarify that this is not a comparison against Chain-of-Thought (CoT), but a demonstration of the **synergy** between TurboConn and CoT. Our results show that when provided with a fixed CoT rationale, TurboConn is more effective at correctly generating the final answer.
>
> ---
>
> ### **Re: Length generalization ability of baseline Transformer**
>
> We conducted additional experiments using training lengths of 20 and 30.
>
> **Training Length 20**: Performance drops to 40.41% when evaluated on a sequence length of 30.
>
> **Training Length 30**: Similarly, accuracy falls to 54.32% when evaluated on a sequence length of 40.
>
> We also observed that as the training sequence length increases, it becomes progressively more difficult for the baseline model to even overfit the training distribution. These results suggest that the standard Transformer architecture lacks the inherent capability for length generalization on this task.
>
> ---
>
>
> ### **Re: Per-token vs. Grouped Connections**
>
> We measured the training time (in seconds) to compare per-token connections to our primary setup (Group Size 4).
>
> | Dataset | Group Size 1 Time (s) | Group Size 4 Time (s) |
> | :--- | :---: | :---: |
> | Parity | 29.7 | 6.4 |
> | Multi-step Arithmetic | 21.4 | 4.9 |
> | GSM8K (No CoT) | 16.2 | 4.0 |
>
> As the data shows, using a group size of 4 significantly improves training performance.
>
> ---
>
> ### **Re: Gradient Stability**
>
> We did not observe issues related to exploding or vanishing gradients during our training process. Theoretically, the inherent attention mechanism creates computational shortcuts across sequence steps, which should help to alleviate these numerical instabilities.
>
> ---
>
> ### **Re: Train from scratch**
> While our primary focus was on fine-tuning pre-trained LLMs, we conducted an additional experiment to test our method when training from scratch. We used a small-scale configuration (8 layers, 512 hidden size, 1024 intermediate size, and 1 attention head) trained on 64,000 samples of the Parity task with 20 digits for 1 epoch.
>
> | Method | Acc (%) |
> | :--- | :---: |
> | Baseline  | 89.61 |
> | **TurboConn** | **99.12** |

---

> > ### Author Rebuttal · Reviewer_WzN3 · 2026-04-03
> >
> > The authors cleared most of my concerns, and I am inclined to raise my score if the remaining concerns (especially the choice of connections) are addressed properly.
> >
> >
> > **Comparison to existing Recurrent-Transformer hybrids.**
> >
> > This was one of my main concerns, and the authors addressed it appropriately. I agree with the authors that their main selling point is the compatibility of existing pre-trained LLMs. Therefore, asking for an experimental comparison with Recurrent-Transformer hybrids is unfair. The authors explained the technical differences in the answer to Reviewer 4mDc. If such a detailed discussion will be included in the final version, I don't see an issue here.
> >
> > **Choice of Connections**
> >
> > Thanks for providing the formula, which shows the hyperparameters involved. Since the hyperparameters are different per model (see Appendix A.4), it does seem that there was some engineering involved to find the optimal wiring. The authors unfortunately evaded the question of "How much effort is required to find the optimal wiring for a given model?" To be a bit more precise here: How robust is the approach to the exact wiring per model? Does performance degrade significantly if a different wiring is chosen? If so, how many configurations were tested until the final wiring was found? I find this relevant to determine how much engineering would be needed to adopt this approach to a new pre-trained model.
> >
> > The soft-token experiment is rather a radical version of an ablation of wiring density. My question was more in the direction of "what happens to the performance for increasing k and l?" I could have phrased this more precisely, and while I would appreciate this experiment out of interest, I don't deem it critical to the paper.
> >
> > **Filter method for GSM8K**
> >
> > Thanks, this is reassuring to hear. Please include this in the paper instead of "overly long".
> >
> > **Chain-of-though**
> >
> > Thanks for the clarification on the distribution shift. It is interesting, though, that such distribution shifts diminish the gains of TurboConn. Do the authors have some conjectures on why? While the experiments do show synergy between TurboConn and CoT, my observation still holds: the advantage of TurboConn is smaller compared to a CoT baseline.
> >
> > **Length generalization**
> >
> > These experiments further strengthen the length generalization findings.
> >
> > **Per-token vs. Grouped Connections**
> >
> > Thanks for the timings, but I was actually asking for evaluations. Sorry for this misunderstanding.
> >
> > **Train from scratch**
> >
> > This is interesting and shows an advantage even when training from scratch. In that sense, the paper might even be competitive to the aforementioned Recurrent-Transformer hybrids. However, I agree that this is not the focus of the paper.

---

> > > ### Author Response · Authors · 2026-04-08
> > >
> > > Thank you so much for your encouraging and insightful comments. We are glad that our responses regarding the comparison to recurrent hybrids, training from scratch experiment, and length generalization were helpful.
> > >
> > > ### **Re: Choice of Connections**
> > > We did not conduct an extensive search for “optimal” wiring. The only decision based on experiment is whether to leave the bottom‑most layers unchanged. They are generally understood to encode basic features rather than the high-level logic targeted by our recurrence. When training appears unstable, we increase $L_{min}$ from 0 to a value that "feels right". This turns out to be sufficient for our method to work.
> > >
> > > The other hyperparameters the connection formula were chosen arbitrarily at the start of our experiments. Our primary constraints were ensuring that the total number of trainable parameters did not exceed the standard baseline and that the connections covered the layers relatively evenly. These configurations were then kept constant across all subsequent experiments.
> > >
> > > Because we conducted one or zero trial per model to determine these hyperparameters, we believe the method is robust to the specific wiring setup, provided the connections are sufficiently dense and balanced across the network depth.
> > >
> > > $$\mathcal{C} = \\{ (s, l) \mid s - l > k, \text{ where } s, l \in [L_{min}, L_{max}] \\text{ sampled at interval } m \\}$$
> > >
> > > ### **Re: Chain of Thought (CoT)**
> > > The distribution shift occurs because the model may learn "pseudo-rules" from the synthetic training data (which contains certain inherent biases) that do not hold true in the general GSM8K test set (e.g., specific arithmetic patterns that only exist when certain numbers are excluded, such as assuming $a \times b \neq b$ if the dataset lacks 0 or 1). TurboConn provides additional capacity to learn these representations. It can diminish the observable gains of TurboConn, as the added reasoning capacity may be diverted toward learning these distribution-specific patterns rather than universal rules. While TurboConn learns a variety of additional internal representations, some of these are specific to the training distribution while others are generally valid. Because only the generally valid representations translate into performance gains on the test set, the observable advantages of our method are naturally more pronounced when the training and testing distributions are better aligned, such as with the Numina dataset.
> > >
> > > ### **Re: Per-token vs. Grouped Connections**
> > > We apologize for the misunderstanding in providing training times rather than performance metrics. Below are the accuracy results comparing per-token connections (Group Size 1) and grouped connections (Group Size 4) using Llama 3.2 1B.
> > >
> > > | Dataset | Group Size 1 (Acc %) | Group Size 4 (Acc %) |
> > > | :--- | :---: | :---: |
> > > | Parity | 100.0 | 100.0 |
> > > | Multi-step Arithmetic | 43.71 | 42.66 |
> > > | GSM8K (No CoT) | 10.60 | 8.32 |

---

### Official Review · Reviewer_4mDc · 2026-03-10

**Soundness:** 3
**Presentation:** 2
**Significance:** 3
**Originality:** 2
**Overall Recommendation:** 3
**Confidence:** 3

**Summary:**

This paper points out that the inference capability of the standard Transformer is limited by a fixed maximum number of steps along the computation path. To address this limitation, the authors propose a novel architecture called Turbo Connection. This architecture breaks through the fixed depth limitation by connecting the high-level hidden state residuals of the previous token t to the low-level layers of the next token t+1. This design leverages the autoregressive properties of Large Language Models, allowing the effective inference depth to grow linearly with the sequence length. Unlike traditional thought chains that require explicit token generation, TurboConn allows the "thinking" process to occur directly within the model's hidden states. Experiments show that, when fine-tuned on pre-trained models (such as Qwen-3-1.7B), TurboConn significantly improves the accuracy of inference benchmarks such as GSM8K, Parity, and multi-step arithmetic without additional generation latency or increased memory consumption.

**Compliance With Llm Reviewing Policy:**

Affirmed.

**Final Justification:**

I sincerely thank the authors for their detailed rebuttal and the subsequent insightful discussion regarding discriminative filtering, mode-seeking behavior, and model reliability. However, the degeneration of the prefill stage into group-sequential processing introduces a non-trivial latency bottleneck, and the necessity of structural fine-tuning raises the barrier to entry.

I believe my current rating accurately reflects both the academic merits and the practical limitations of this work. Therefore, my score remains unchanged.

**Key Questions For Authors:**

1. Since TurboConn is fine-tuned and evaluated on a pre-trained model, how are the newly introduced cross-layer residual connections initialized (e.g., are they zero-initialized to perfectly preserve the distribution characteristics of the pre-trained model )? How sensitive is the model to this initialization in the early stages of fine-tuning?

2. In the standard Transformer's Prefill phase, all tokens are computed in highly parallel. If the connections from higher layers at t to lower layers at t+1 are strictly followed, does this mean the Prefill phase must degenerate into serial computation or require a special chunking attention mechanism?

3. The experiments primarily demonstrate a significant improvement over Qwen-3-1.7B. Do the authors have any preliminary theoretical intuition or small-scale ablation experiments indicating that this improvement can be generalized to extremely deep, state-of-the-art models? For models that already have sufficient depth, does this additional "temporal depth" still provide marginal benefits?

**Limitations:**

yes

**Strengths And Weaknesses:**

1. Soundness:

Strengths: The paper's empirical evaluation covers a variety of inference tasks (such as Parity, GSM8K, and multi-step arithmetic), providing strong performance proofs. The authors also conducted ablation experiments on connection "density," demonstrating the significant advantage of dense back-connections over sparse connections, showcasing the rigor of the research.

Weaknesses: The evaluation primarily focuses on the model's fine-tuning phase. Although the authors claim compatibility with existing pre-training objectives, resource constraints prevent empirical evidence of pre-training the architecture from scratch. Additionally, current experiments are primarily conducted on smaller models (such as Qwen-3-1.7B), and it remains uncertain whether such improvements will maintain equivalent scaling laws on larger models with hundreds of billions of parameters.

2. Presentation

Strengths: The core motivation is intuitive and clear: it transforms fixed-depth computation into time-varying depth computation through a connection mechanism of  t  to t+1. The paper effectively places this method within the context of "latent reasoning," clearly distinguishing the difference between information flow within hidden states and explicit token projection.

Weaknesses: The lack of extremely detailed pseudocode for tensor shape transformation and forward propagation in the paper will significantly hinder readers' ability to reproduce the method. It is recommended that the authors supplement the appendix with specific PyTorch pseudocode implementations to improve reproducibility.

3. Significance

Strengths: TurboConn completely retains the standard training objective which significantly enhances its potential for deployment.

Weaknesses: Due to the need to modify the Transformer's underlying architecture, structural weight fine-tuning is required, which increases the barrier to entry to some extent.

4. Originality

Strengths: The paper proposes a highly innovative new perspective by routing information from the "higher level" of the current token to the "lower level" of the next token to overcome the depth bottleneck of Transformers.

Weaknesses: Conceptually, this hierarchical connection across time steps bears some resemblance to earlier attempts to introduce loop mechanisms into Transformers (such as Universal Transformers or architectures with cross-token state preservation). The paper needs to further demonstrate, theoretically and mechanistically, the fundamental difference between its routing mechanism and these traditional RNNs or recurrent Transformers in handling information bottlenecks.

---

> ### Author Rebuttal · Authors · 2026-03-30
>
> We sincerely appreciate Reviewer 4mDc for the insightful review. We appreciate the opportunity to provide a more detailed analysis of our results.
>
> ---
>
> ### **Re: Include pseudocode implementation, change in tensor shape**
>
> To ensure full reproducibility of TurboConn, we will provide the following detailed pseudocode for the forward pass in the appendix: https://osf.io/xgrh2/overview?view_only=277783a27533493bac47c7d1a69c19c6
> Furthermore, we would like to highlight that we have submitted our complete source code as supplementary material, including the modified model definitions for Llama and Qwen, which allows for direct execution and verification of our results.
>
> ---
>
> ### **Re: Due to the need to modify the Transformer's underlying architecture, structural weight fine-tuning is required, which increases the barrier to entry to some extent.**
>
> We recognize that introducing TurboConn necessitates additional training. While our method requires structural fine-tuning, it crucially allows us to leverage the vast representations already learned by the LLM during pre-training, avoiding the need to train from scratch.
>
> ---
>
> ### **Re: Demonstrate the fundamental difference between its routing mechanism and these traditional RNNs or recurrent Transformers in handling information bottlenecks.**
>
> While both TurboConn and the Universal Transformer utilize recurrence to increase depth, Universal Transformer is recurrent in depth (per token), whereas TurboConn is recurrent across the sequence (cross-token). To clarify the mechanical differences, we compare the two architectures under the assumption that both are configured to increase effective reasoning depth by a factor of $D$:
>
> **1. Computational Efficiency and Scaling**
>
> To increase the effective reasoning depth by a factor of $D$, the Universal Transformer must perform $D$ recursive iterations for **every token** in the sequence.
> * **Universal Transformer:** Training and inference time scale linearly with $D$ ($O(D)$ complexity per token).
> * **TurboConn:** TurboConn scales depth by utilizing the sequence dimension. While it breaks full token-level parallelism, it does not multiply the total FLOPs per token. Because it preserves intra-group and tensor-wise parallelism, the training time increases by a factor significantly less than $D$.
>
> **2. Memory Footprint**
> * **Universal Transformer:** To achieve a depth factor of $D$, the memory used should grow by a factor of $D$ because the number of states that must be stored for each recursion increases.
> * **TurboConn:** TurboConn introduces no significant change to the memory cost; the existing hidden states are sufficient to capture and convey the required information when backpropagating through backward connections.
>
> **3. Autoregressive Generation and Inference**
>
> The difference is also pronounced during the autoregressive generation phase.
> * **Universal Transformer:** Each generated token requires $D$ full passes through the recurrent layers, increasing latency by a factor of $D$.
> * **TurboConn:** TurboConn involves little additional cost during generation. The backward connections are integrated into the single forward pass of the model, allowing for depth-scaling without the linear latency penalty of traditional recurrent Transformers.
>
> ---
>
> ### **Re: Initialization Strategy**
> As mentioned in Section 3.2, we utilize zero-initialization for the bridge layers so that the model's behavior is identical to the original pre-trained LLM at the start of training. In our preliminary tests, we found that without zero-initialization, the initial loss is extremely high; we believe this introduces unnecessary instability that provides no benefit to the subsequent finetuning process.
>
> ---
>
> ### **Re: If the connections from higher layers at t to lower layers at t+1 are strictly followed, does this mean the Prefill phase must degenerate into serial computation or require a special chunking attention mechanism?**
> Yes, during the prefill phase, tokens are processed sequentially at the group level, while maintaining full parallelism within each group. This introduces a configurable trade-off: a larger group size facilitates parallel execution and reduces latency, whereas a smaller group size provides greater reasoning depth, which may be beneficial for complex instructions.
>
>
> ---
>
> ### **Re: Implications for larger models**
>
> TurboConn provides a qualitative advantage—Discriminative Filtering—that scaling alone does not replicate. In our experiments, the 1.7B model with TurboConn eliminated significantly more mathematically "impossible" digit choices (5.259) than the much larger 8B baseline (4.500), despite both achieving nearly identical accuracy (~45%). This suggests that our "temporal depth" creates an internal verification mechanism that can potentially reduce hallucinations by preventing the model from assigning probability to "surely wrong" answers.

---

> > ### Author Rebuttal · Reviewer_4mDc · 2026-04-02
> >
> > I would like to thank the authors for their highly detailed response. However, after carefully re-evaluating the manuscript and the rebuttal, I have decided to maintain my original score. While the authors successfully clarified how the architecture works, the explanations essentially confirm the significant inherent trade-offs of the proposed method:
> >
> > 1. Prefill Stage Serialization: The authors confirmed that the prefill stage degenerates to group-sequential processing. In the context of modern LLM deployment, where processing long contexts efficiently is critical, sacrificing full token-level parallelism introduces a non-trivial latency bottleneck. The "configurable trade-off" between group size and inference depth remains a substantial practical limitation.
> >
> > 2. Performance and Architectural Complexity: Regarding the impact on larger models, while the "discriminative filtering" observation on the 1.7B model is intriguing, the rebuttal notes that its final accuracy (~45%) is nearly identical to the 8B baseline. This suggests that despite the significant architectural modifications (and the barrier of structural fine-tuning), the end-to-end downstream performance does not demonstrate a compelling advantage over simply scaling up the model.
> >
> > In summary, the rebuttal effectively resolved my ambiguities, but it also reinforced my assessment that the method comes with heavy architectural compromises (prefill latency and structural fine-tuning) that limit its practical impact. The paper is a interesting exploratory study, but I think my current rating accurately reflects its contributions and inherent limitations.

---

> > > ### Author Response · Authors · 2026-04-07
> > >
> > > We thank Reviewer 4mDc for their thoughtful response and for finding
> > >
> > > * our experiment showcasing rigor of the research
> > >
> > > * our work proposing a highly innovative perspective
> > >
> > > * and the emergence of discriminative filtering ability of TurboConn to be intriguing
> > >
> > >
> > > Regarding "end-to-end downstream performance," we argue that top-1 accuracy is an incomplete metric for reasoning. As noted by Huszár (2015), standard maximum likelihood training optimizes the forward KL divergence ($KL(P\|Q)$). A known property of forward KL is that it is "mean-seeking"; the model ($Q$) tries to cover all modes of the data ($P$). This ensures the model doesn't "miss" the correct answer, but it often leads to a distribution that is too broad, assigning probability mass to regions where the data has zero density. This is why models "have a tendency to produce completely unseen sequences."
> > >
> > >
> > >
> > > In contrast, a model that minimizes reverse KL ($KL(Q\|P)$) is "mode-seeking" and prefers to assign zero probability to anything implausible, even if it misses some correct modes. While accuracy only measures if the correct answer is the mode, reliability depends on the model’s ability to "zero out" mathematically impossible paths.
> > >
> > >
> > >
> > > Interestingly, our method shows emergent ability to better filter out implausible choices without requiring expensive alternative objectives. We believe this internal verification ability is a critical dimension of performance, especially as model reliability becomes a primary concern in the field.
> > >
> > > ---
> > > **References**
> > > Huszár, 2015, How (not) to Train your Generative Model: Scheduled Sampling, Likelihood, Adversary? arXiv:1511.05101.

---

### Official Review · Reviewer_ehJg · 2026-03-11

**Soundness:** 2
**Presentation:** 3
**Significance:** 3
**Originality:** 3
**Overall Recommendation:** 4
**Confidence:** 3

**Summary:**

The authors suggest augmenting the Transformer architecture with higher-to-lower layer connections. Specifically, they pass the hidden state of a higher layer to a lower layer at the next token. They demonstrate that this results in improved accuracy on mathematical and logical tasks.

**Compliance With Llm Reviewing Policy:**

Affirmed.

**Key Questions For Authors:**

* Do these gains transfer to more challenging tasks? For example, could you train the model on the OpenThoughts dataset and evaluate it on recent math benchmarks (AIME24/25, AMC, MATH-500), or test other domains as well? What about general LM abilities?
* Please show how model performance compares between the architectures at the same computational budget throughout training, similar to Figure 1 in Gerasimov et al., “You Do Not Fully Utilize Transformer’s Representation Capacity.” Also, what is the inference-time overhead from the additional computation?
* Is the “Baseline” an SFT-trained vanilla Transformer or an untrained model? Please, add both. Also, please report the confidence intervals or standard deviations.
* How are the connection locations chosen?
* It is interesting that Llama-1B shows much better performance on the parity task than the larger Qwen model. Is this expected?

**Limitations:**

yes

**Strengths And Weaknesses:**

**Soundness**

The authors’ claims are generally well supported by experimental evidence, with no major problems in the setup. The main weaknesses are the simplicity of the selected tasks and the lack of a fair comparison at the same training budget.

**Presentation**

The paper is well written and easy to follow, though some experimental details are missing or hard to find.

**Significance**

It is somewhat modest, since the authors suggest only a marginal change to the Transformer architecture. Still, it is an interesting idea that may be valuable to the community.

**Originality**

To the best of my knowledge, the architecture itself is novel and is well articulated in terms of how it builds on prior work.

---

> ### Author Rebuttal · Authors · 2026-03-30
>
> We sincerely thank Reviewer ehJg for their insightful feedback. Below we address your follow-up questions in detail.
>
> ---
>
> ### **Re: Do these gains transfer to more challenging tasks?**
> We have launched an experiment to see how well our method generalize to more recent math benchmarks. Due to the computational cost, we could not complete this within the rebuttal periods, but commit to including the results in the final version.
> Given that current datasets consistently demonstrate that increased depth yields performance gains under identical training setups, we expect this principle to hold on more recent datasets as well.
>
> ---
>
> ### **Re: Compare at the same computational budget**
>
> We agree that it is important to evaluate model performance with respect to the computational budget. We provide a detailed analysis of FLOPs, which serves as the standard measure for computational budget in many architectural evaluations, such as in Figure 1 of Gerasimov et al. (2025).
>
> Because the bridges we introduce are low-rank and do not modify the existing Transformer layers, the overall impact on the total FLOP budget is minimal.
>
> **Example Calculation: Llama 3.2 1B**
>
> 1.  **Baseline FFN FLOPs:** For a sequence length $T$ and hidden size $d$, each FFN layer (comprising gate, up, and down projections) performs $6 \cdot T \cdot d \cdot d_{ff}$ FLOPs.
>
> 2.  **TurboConn FLOPs:** Each bridge (down and up projections) performs $4 \cdot T \cdot d \cdot r$ FLOPs.
>
> 3.  **Relative Overhead:** Comparing the bridge overhead to the FFN computation yields the following expression:
>     $$\text{Relative Overhead} = \frac{B \cdot 4 \cdot T \cdot d \cdot r}{L \cdot 6 \cdot T \cdot d \cdot d_{ff}} = \frac{2 \cdot B \cdot r}{3 \cdot L \cdot d_{ff}} = \frac{2 \cdot 15 \cdot 120}{3 \cdot 16 \cdot 8192} \approx \mathbf{0.91\%}$$
>
> Even when compared strictly to the FFN layers—and excluding the additional FLOPs from Attention and Embeddings—the overhead is insignificant. This applies equally to the backward pass during training, which follows the same logic.
>
> **Inference-Time Overhead**
>
> Regarding the overhead during inference, the logic remains the same: the addition of low-rank bridges results in a negligible increase in total computation. Furthermore, the break in parallelism required for TurboConn training no longer has an effect during the autoregressive generation phase. Because autoregressive inference is inherently sequential—processing one token at a time—the "recurrent" nature of TurboConn aligns perfectly with the existing generation bottleneck.
>
> ---
>
> ### **Re: Baseline Definition**
> As detailed in Sections 4.2 and 4.3, **"Baseline"** results refer to Qwen and Llama finetuned under the exact training setup as the TurboConn variant. We have updated the table below to include **Zero-shot** performance (the original model).
>
> | Model | Dataset | Method | Acc (%) |
> | :--- | :--- | :--- | :--- |
> | **Llama 3.2 1B** | GSM8K (No CoT) | Zero-shot | 0.84 |
> | | Multi-step Arithmetic | Zero-shot | 4.15 |
> | | Parity | Zero-shot | 30.16 |
> | **Llama 3.1 8B** | GSM8K (No CoT) | Zero-shot | 3.69 |
> | | Multi-step Arithmetic | Zero-shot | 10.49 |
> | | Parity | Zero-shot | 42.28 |
> | **Qwen 3 1.7B** | GSM8K (No CoT) | Zero-shot | 10.50 |
> | | Multi-step Arithmetic | Zero-shot | 19.40 |
> | | Parity | Zero-shot | 51.40 |
>
> Regarding statistical variance, our evaluation is deterministic as it calculates the exact log-likelihood of the ground-truth answers across the full test set. Since we are not sampling, the results represent the definitive performance of the model on these benchmarks.
>
> ---
>
> ### **Re: Choice of connection locations**
> The specific wiring configurations were derived from a simple heuristic rather than exhaustive optimization. Our principle was to ensure broad layer coverage while maintaining a low computational overhead. For each model, we applied a rule-based pattern:
> $$\mathcal{C} = \\{ (s, l) \mid s - l > k, \text{ where } s, l \in [L_{min}, L_{max}] \\text{ sampled at interval } m \\}$$
> We intentionally leave the bottom-most layers unchanged, as they are generally understood to encode basic features rather than the high-level logic targeted by our recurrence.
>
> ---
>
> ### **Re: It is interesting that Llama-1B shows much better performance on the parity task than the larger Qwen model. Is this expected?**
> While it may seem counterintuitive that the smaller Llama-1B outperforms the larger Qwen model on the Parity task, this highlights that raw parameter count is not the sole determinant of reasoning capability. Our observations suggest that Llama’s architectural configuration—specifically its wider FFN intermediate layers—provides a more favorable inductive bias for the 'accumulator' logic required by Parity. Qwen’s deeper but narrower design may prioritize sequential refinement that, within the constraints of a finite, non-recurrent forward pass, cannot fully compensate for its reduced per-layer capacity.

---

### Decision · Program_Chairs · 2026-04-30

**Decision:**

Accept (regular)

**Comment:**

The paper shows that the reasoning limitations of Transformers (caused by a fixed-depth computational path) can be mitigated using a novel architecture called Turbo Connection (TurboConn). This method routes residual connections from higher-layer hidden states of a token to the lower layers of the subsequent token, dynamically extending the computational depth without heavily impacting generation latency.

## Comments

On the positive side, the reviewers found that this is an elegant, innovative approach that retains training objectives while yielding accuracy gains on benchmarks like GSM8K, Parity, and multi-step arithmetic. Reviewers liked the compatibility with existing pre-trained models, the nice ablation studies (e.g., why dense connections are important), and the new findings regarding length generalization.

The reviewers identified areas of improvement:

- It would be best to eval on new math reasoning benchmarks to prove the method's robustness.

- Directly address the latency bottleneck introduced by group-sequential processing during the prefill stage.

- Replace the overly abstract diagrams with precise architectural visualizations

- Expand the discussion comparing the computational trade-offs of this method against existing Recurrent-Transformer hybrids.

Please integrate this and other discussion points into the revision of this paper.

## Recommendation

Based on these reviews, I recommend accepting this paper. The reviewers liked he paper's perspective on latent reasoning and the emergent "discriminative filtering" capability, which enhances model reliability by suppressing impossible paths. The validation of increasing temporal depth to get better reasoning capabilities presents a valuable contribution to new models of LLMs.